# Patient Preferences in Metastatic Breast Cancer Care: A Scoping Review

**DOI:** 10.3390/cancers15174331

**Published:** 2023-08-30

**Authors:** Kelcey A. Bland, Reem Mustafa, Helen McTaggart-Cowan

**Affiliations:** 1Cancer Control Research, BC Cancer Research Institute, Vancouver, BC V5Z 1L3, Canada; kelcey.bland@ubc.ca (K.A.B.); rmustafa@bccrc.ca (R.M.); 2Department of Physical Therapy, University of British Columbia, Vancouver, BC V6T 1Z4, Canada; 3Faculty of Health Sciences, Simon Fraser University, Burnaby, BC V5A 1S6, Canada

**Keywords:** breast cancer, conjoint analysis, discrete choice experiment, healthcare provider preferences, metastases, oncology, patient preferences, preference elicitation, qualitative research, review

## Abstract

**Simple Summary:**

Identifying and understanding patient preferences regarding their own care can help to tailor cancer therapies and services to the needs, goals, and values of patients. Currently, research on the preferences of patients with metastatic breast cancer (MBC) regarding their care has not been summarized. Our review aims to summarize all research reporting on the specific preferences of patients with MBC regarding their care to identify important areas for future research. The main finding of the current review is that to-date studies evaluating preferences among patients with MBC are mixed. Most studies on MBC patient preferences have focused on capturing preferences directly relating to cancer treatments. More information on patient preferences for other aspects of MBC care, including supportive care therapies and services that target physical, mental, and emotional quality of life, is needed.

**Abstract:**

People with metastatic breast cancer (MBC) have diverse medical, physical, and psychosocial needs that require multidimensional care. Understanding patient preferences is crucial to tailor treatments, services, and foster patient-centered care. A scoping review was performed to summarize the current evidence on the preferences of people with MBC regarding their care to identify knowledge gaps and key areas for future research. The Embase, MEDLINE, CINAHL and PsycInfo databases were searched. Twenty studies enrolling 3354 patients met the study eligibility criteria. Thirteen quantitative studies, four mixed methods studies, and three qualitative studies were included. Seven studies captured healthcare provider perspectives; thirteen studies evaluated patient preferences relating specifically to cancer treatments; three studies evaluated preferences relating to supportive care; and four studies evaluated communication and decision-making preferences. The current literature evaluating MBC patient preferences is heterogeneous with a focus on cancer treatments. Future research should explore patient preferences relating to multidisciplinary, multi-modal care that aims to improve quality of life. Understanding MBC patient preferences regarding their comprehensive care can help tailor healthcare delivery, enhance the patient experience, and improve outcomes.

## 1. Introduction

Breast cancer is a widespread global health issue, with a high incidence rate and ranking second among cancer-related deaths in North America [1]. Metastatic breast cancer (MBC), where cancer cells have spread beyond the breast and surrounding lymph nodes, is responsible for a significant proportion of breast cancer deaths worldwide [2]. While only 6% of new breast cancer diagnoses are MBC, 30% of people with early-stage breast cancer develop MBC later in life [1]. Unlike early-stage breast cancer, which is treated with curative intent, MBC is incurable and necessitates multiple rounds of systemic cancer therapies to manage the disease, prevent or delay progression, and maintain quality of life [2,3]. While current therapies have extended the average life expectancy of people living with MBC, the adverse effects of treatment coupled with the symptoms of MBC itself, impose a significant physical and psychosocial burden on patients [4,5,6,7]. Consequently, there is a need for patient-centered multidimensional care to address the diverse needs of people with MBC.

Identifying and understanding patient preferences is a crucial step towards fostering patient-centered care and helps to tailor MBC therapies to the needs, goals, and values of patients. Evaluating patient preferences can also help healthcare providers (HCPs) identify areas for improvement in the delivery of care and facilitate shared decision-making [8]. Patient preferences can be identified both quantitatively (e.g., discrete choice experiments (DCEs), conjoint analysis) and qualitatively (e.g., patient interviews, focus groups). Conjoint analysis involves presenting patients with a series of hypothetical treatment scenarios, each with multiple attributes, and asking them to rate or rank the importance of each attribute. The analysis of the ratings or rankings allows researchers to determine the relative importance of each attribute and how different combinations of attributes affect overall preferences [9]. DCEs are a choice-based conjoint analysis. DCEs involve presenting patients with hypothetical treatment scenarios and asking them to choose between different options based on various attributes, such as side effects, efficacy, route of administration, and cost [10]. Randomized controlled trials (RCTs) and cross-sectional or longitudinal observational study designs may also include patient preferences on specific interventions as an outcome measure, although cannot provide information on trade-offs between different attributes. Alternatively, qualitative research aims to generate richer, and more detailed insight into patients’ preferences by inquiring about complex patient experiences and perspectives. Ideally, both quantitative and qualitative research methods are needed to fully capture the depth and breadth of patient preferences to ultimately improve healthcare.

To our knowledge, the specific preferences of people with MBC regarding their multidimensional care, from systemic treatments to palliative and supportive care, have not been summarized. Guerra et al. summarized treatment preferences for breast cancer evaluated using DCEs that enrolled people with early-stage or MBC [11]. Other reviews have solely focused on the preferences of people with early-stage breast cancer [12,13,14,15]. Differences in therapies and symptom burden between early-stage and MBC may mean that people with MBC have unique preferences. People with early-stage breast cancer, for example, might be more likely to prioritize treatments with a higher chance of cure, while people with MBC may prioritize quality of life and delaying disease progression [16]. Understanding the preferences of people with MBC regarding their treatment as well as other aspects of care can help inform treatment decisions, improve patient outcomes, and enhance the patient experience. The aim of the current scoping review was to map the available quantitative and qualitative evidence reporting on patient preferences in MBC care and identify relevant knowledge gaps to inform directions for future research.

## 2. Materials and Methods

### 2.1. Search Strategy

A comprehensive search of Embase, MEDLINE, CINAHL and PsycInfo databases was performed. The search strategy was developed in consultation with a research librarian at the BC Cancer Research Institute. Example search terms include “breast neoplasm” and “metastasis/advanced/stage IV” or “palliative/end-of-life” and “patient preference/patient satisfaction.” Search results were restricted to the English language and from 1 January 2000 to 24 October 2022.

### 2.2. Inclusion and Exclusion Criteria

Article eligibility criteria included (1) quantitative, qualitative, or mixed research methods studies reporting on patient preferences regarding any aspect of MBC care (e.g., treatment, supportive care, palliative care, communication) and (2) studies that solely enrolled people with MBC or people with advanced breast cancer and MBC (e.g., stage III–IV). The decision to include people with advanced breast cancer was made, as studies frequently include people with either advanced breast cancer or MBC. Excluding any study that also enrolled people with advanced breast cancer would mean missing important preferences of people with MBC. Exclusion criteria included (1) studies that included people with early-stage cancer or patients with mixed metastatic cancer types (i.e., cancers other than breast), and (2) review papers, conference abstracts, and letters to the editor or commentaries.

Article eligibility was independently reviewed by three authors (K.A.B, R.M., H.M.-C.) using an online software system (Rayyan, https://www.rayyan.ai/). Titles and abstracts were independently screened by at least two blinded reviewers. K.A.B. reviewed all titles and abstracts, and R.M. and H.M.-C. each reviewed 50%. Relevant full-text versions of papers were reviewed to determine eligibility independently by at least two reviewers and data extraction was completed in duplicate. Discrepancies were discussed among all authors until agreement was reached.

### 2.3. Data Extraction and Analysis

Data extraction included: year, authors, country, study design or methodology, patient population, HCP population (if included), sample size, patient age, de novo/recurrent status, preference type (e.g., treatment), outcomes relating to preferences, follow-up, and preference results. A narrative synthesis of the studies was conducted in line with the scoping review’s aim and due to the wide heterogeneity of articles.

## 3. Results

A total of 873 articles were identified through the database search. Following the removal of duplicates, 733 articles were screened and 20 studies with a total of *n* = 3354 patient participants and *n* = 436 HCPs met the eligibility criteria to be included in the review (Figure 1). Articles included were quantitative studies, namely RCTs (*n* = 3) [17,18,19], DCEs (*n* = 3) [20,21,22], conjoint analysis (*n* = 5) [23,24,25,26,27], and other observational cross-sectional or longitudinal designs (*n* = 2) [28,29], mixed research methods studies (*n* = 4) [30,31,32,33], or qualitative studies (*n* = 3) [34,35,36]. Most studies enrolled people with MBC (*n* = 16) [18,19,21,23,24,25,26,28,29,30,31,32,33,34,35,36]. There were four studies that enrolled people with locally advanced and MBC [17,20,22,27]. HCP preferences or perspectives were evaluated in seven studies [17,18,20,21,32,34,35]. Study characteristics are summarized in Table 1.

### 3.1. Treatment Preferences

Most studies evaluated preferences relating to MBC treatment [17,18,19,20,21,22,23,24,25,26,27,28,30], which is one aspect of MBC multidimensional care. Treatment preferences were explored in 12 quantitative studies [17,18,19,20,21,22,23,24,25,26,27,28] and one mixed research methods study [30]. One study explored specific drug preferences [17]. Decker et al. evaluated patient preferences for everolimus in combination with exemestane or capecitabine in combination with bevacizumab for locally recurrent or inoperable MBC in an RCT [17]. No statistically significant differences in preferences for the evaluated treatments were found, although both patients and HCPs tended to favour capecitabine in combination with bevacizumab due to improved quality of life [17].

Four studies evaluated treatment administration (e.g., mode of treatment delivery) preferences [18,19,28,30]. Gornas and Szczylik evaluated patient preferences for intravenous versus oral administration of capecitabine [28]. All participants who completed the survey preferred oral capecitabine, citing increased convenience as the primary reason (72%). Ciruelos et al. and Pivot et al. evaluated patient and HCP preferences for intravenous versus subcutaneous trastuzumab administration [18,19]. Ciruelos et al. reported that patient participants (*n* = 137, 86.2%) as well as HCPs (*n* = 34, 87.2%) preferred subcutaneous versus intravenous administration [18]. Patients preferred subcutaneous administration via a single injection device (*n* = 90, 59.2%) compared to a single injection vial (*n* = 40, 26.3%) [18]. Pivot et al. also found subcutaneous versus intravenous administration was preferred by patients (*n* = 79, 84.9%) and that HCPs were most satisfied with subcutaneous administration (*n* = 56, 88%) [19]. Fallowfield et al. explored patient preferences for intravenous versus oral bisphosphonate administration and preferences for bisphosphonate treatment regimens [30]. The preferred treatment regime for oral therapy was one tablet per month (*n* = 13, 35%) compared to daily or weekly tablets. At 6 months of oral treatment, eight patients (22%) said that they would prefer intravenous instead. Alternatively, six patients (26%) on intravenous treatment said they would prefer oral therapy instead [30]. The preferred treatment regime for intravenous therapy was 15 min every 4 weeks (*n* = 10, 44%) compared with infusions for 2–4 h every 4 weeks or every 3 months.

Eight studies evaluated preferences for treatment characteristics (e.g., treatment efficacy, side-effects) [20,21,22,23,24,25,26,27]. All studies explored the characteristics of tumour-directed treatments prescribed with the intention to slow or delay the progression of MBC. Across four studies, improved treatment efficacy (e.g., prolonged overall or progression-free survival) was most often preferred by patients [20,22,23,25]. Amin et al. reported that patients and HCPs (oncologists) most valued improving overall survival in regard to HER2- treatments for advanced breast cancer, followed by avoiding nausea and vomiting [20]. Mansfield et al. also evaluated HER2- treatment characteristic preferences and found improving progression-free survival from 5 to 26 months was most important to patients [22]. Reducing the risk of heart failure by 15%, reducing the risk of serious lung infections by 15%, and avoiding the possibility of severe liver functions were the next most important characteristics [22]. DaCosta DiBonaventura et al. evaluated preferences for the treatment of MBC and found overall survival was most important to patients followed by avoiding side effects (alopecia and fatigue being most important), and then dosing regimen [23]. Smith et al. evaluated preferences for paclitaxel and capecitabine characteristics among people with MBC [25]. Patients preferred to undergo treatment with a 27–33% likelihood of benefit (i.e., shrinkage of advanced cancer, responding to treatment) regardless of the toxicity scenario presented [25].

In two studies, patients preferred to avoid treatment side-effects or to maintain quality of life relative to improving overall or progression-free survival [26,27]. Spaich et al. evaluated taxane chemotherapy preferences and found avoiding clinically significant neutropenia was most important to people with MBC (relative importance score (RIS): 20.35), followed by avoiding alopecia (RIS: 18.02) and severe neuropathy (RIS: 16.79) [26]. Progression-free survival benefit was fourth most important (RIS: 14.56) [26]. Reinisch et al. evaluated palliative treatment preferences in people with locally advanced or MBC and reported that quality of life (defined as physical agility and mobility) was most important (utility score: 19.4 of 100%), followed by overall survival (score: 15.2%) and progression free survival (score: 14.4%) [27].

There were two studies that did not include treatment efficacy as a characteristic and focused on preferences to avoid specific treatment side-effects [21,24]. Lalla et al. evaluated patient treatment preferences and willingness to pay for MBC treatment to avoid side effects [24]. People with MBC were willing to pay the most, USD 3894, to avoid severe diarrhea, followed by USD 3479 to avoid being hospitalized due to infection, USD 3211 to avoid severe nausea, USD 2764 to avoid severe tingling in hands and feet, USD 2652 to avoid severe fatigue, USD 1853 to avoid obvious hair loss, and USD 1458 to avoid severe pain [24]. Treatment attributes that were most important to people with MBC in terms of average utility were risk of infection, diarrhea, and nausea [24]. Maculaitis et al. explored patient preferences for CDK4/6 inhibitor regimens for HR+ and HER2- advanced or MBC and found lowering the risk of diarrhea/severe diarrhea and Grade ¾ neutropenia/serious infection were most important to both patients and HCPs (oncologists) [21].

### 3.2. Communication and Decision-Making Preferences

Four studies evaluated communication and decision-making preferences [31,32,34,35], typically with a focus on preferences for treatment information and decision-making. Two qualitative studies evaluated communication preferences regarding treatment decision-making and information as well as prognosis among people with MBC and their HCPs [34,35]. Niranjan et al. conducted interviews with patients and focus groups with HCPs to explore communication preferences regarding prognosis, crisis support, treatment information, and communication timing [35]. Five themes were generated: (1) most patients wanted prognostic information but differed in the timing of when they wanted to have this conversation; (2) emotional distress and discomfort was a critical reason for not discussing prognosis; (3) religious beliefs shaped preferences for prognostic information; (4) HCPs differed on prognostic information delivery timing; and 5) HCPs acknowledged that an individualized approach taking into account patient values and preferences would be most beneficial. Butow et al. investigated the experiences and attitudes towards discussing prognosis, views about the ideal content, and process of information delivery about cancer through semi-structured interviews with patients and HCPs [34]. Seven themes were generated, two related to patient preferences: (1) open and repeated negotiations for patient preferences for information and (2) clear, straight-forward presentation of prognosis where desired [34]. Patients expressed that HCPs should not assume patients want to be told their prognosis [34]. HCPs emphasized it is critical to assess patients’ preferred information levels and be sensitive to changes [34].

Two mixed methods studies evaluated treatment decision-making preferences [31,32]. Ejem et al. evaluated decision-making preferences overtime among women with MBC using the Control Preferences Scale (CPS) and semi-structured interviews [31]. The authors reported incongruence between the CPS and interview findings at baseline (32% congruence, kappa = 0.083) and 3 months (33% congruence, kappa = 0.120). CPS results suggested most patients preferred shared treatment decision-making at baseline (*n* = 14, 64%) and 3 months (*n* = 9, 47%) [31]. However, interviews revealed that with actual experiences of decision-making, patients typically delegate treatment decision-making to their oncologist. Rocque et al. investigated treatment decision-making preferences using the CPS and interviews with patients and focus groups with HCPs [32]. Three themes were generated relating to decision-making preferences: (1) preferences may be influenced by treatment choice; (2) contextual factors set the stage for the decision and influence preferences; and (3) decision-making style provides a baseline approach to decision-making. Relating to the third theme, CPS results indicated >45% of patients and 60% of HCPs preferred shared decision-making. Patient interviews indicated decision-making styles were influenced by personal characteristics, spirituality, and inclusion of others.

### 3.3. Supportive Care Preferences

The remaining three studies explored supportive care preferences [29,33,36]. Delrieu et al. explored physical activity preferences, including physical activity counseling source, location, and modality, as part of secondary analyses within a single-arm physical activity intervention trial [29]. Patient preferences were captured at baseline and after the 6-month intervention among people with MBC. Top preferences included physical activity counseling with a physical activity specialist (baseline: *n* = 30, 65.2%, 6 months: *n* = 34, 81.0%), physical activity delivered at a cancer center (baseline: *n* = 21; 47.8%, 6 months: *n* = 14, 34.1%), and face-to-face settings (baseline: *n* = 37, 82.2%, 6 months, *n* = 30, 73.2%) [29]. Preferences for physical activity program features, such as timing (pre, during and post-treatment), intensity, level of supervision, were also explored [29]. Ten Tusseh et al. assessed exercise-based physical therapy programming preferences for people with MBC in a mixed methods study via surveys and a series of focus groups [33]. Survey-based results indicated that patient preferences varied. However, top preferences included being “active in own environment (walking, cycling, swimming, etc.)” (*n* = 60, 53%) and “fitness training (endurance)” (*n* = 51, 45%). Most patients also preferred being active in a group (*n* = 33, 29%) or individually (*n* = 31, 27%). More than half of all patients expressed wanting exercise-based physical therapy programming with at least weekly supervision of a physical therapist. Focus group data clarified that patients had a particular interest in programs supervised by qualified physical therapists, given many patients feel insecure about being able to self-manage their physical function [33]. Other specific exercise-based program or training features were evaluated including exercise frequency, intensity, and program duration. Finally, in order to assess self-management preferences, experiences and practices, Schulman-Green et al. conducted a survey of people with MBC [36]. Three themes were generated: (1) self-management practices (relate to caring for health, and communication with family and friends, and with HCPs; (2) preferences for participation in self-management range (from passive to active); and (3) facilitators and barriers to self-management [36].

## 4. Discussion

The main finding of the current scoping review is that studies to-date evaluating preferences among people with MBC are heterogeneous. However, most research on MBC patient preferences is restricted to eliciting preferences relating to treatments or treatment communication and decision-making. Only a small proportion of studies included in our review explored patient preferences for other aspects of MBC care, such as physical activity, self-management practices, and communication about prognosis. HCP views were captured in seven studies, of which four focused on treatment preferences and three focused on treatment or prognosis communication and decision-making preferences. No studies on supportive care preferences captured the perspectives of HCPs.

Studies evaluating treatment preferences in our review largely focused on preferences for tumour-directed treatment. We did not identify any studies that explored patient preferences for treatments that focused on managing symptoms arising from the cancer or treatments, such as anxiety or depression, sleep disturbances, or pain [4,5,6,7], other than Fallowfield et al., who explored preferences on the mode of bisphosphonate treatment administration [30]. There is continued research on the medical management of symptom burden among people with MBC [37] and more information on patient preferences for such treatments is needed. Most studies in our review explored which tumour-directed treatment characteristics were most important to patients using DCEs or conjoint analyses (62%, 8 out of 13 treatment preference studies). It was often reported that overall survival, progression-free survival, or superior treatment benefit, were most important to people with MBC relative to managing treatment side-effects or other treatment characteristics (such as dosing regimen). The tendency for people with MBC to prefer greater treatment benefit and to trade-off risk of side effects has also been found in other studies, including people with earlier-stage breast cancer [12,38]. Across all studies evaluating treatment characteristic preferences in our review, preserving quality of life was not often included as an outcome. Only one study included quality of life as an outcome (defined as preserving physical agility and mobility) and reported that preserving quality of life was most important to patients relative to extending both overall and progression-free survival as well as avoiding treatment side-effects [27]. Consequently, while both patients and HCPs tend to consider treatment benefit and efficacy (e.g., extending survival) and mitigating side effects as important, information on the relative importance of maintaining physical function (i.e., ability to perform activities of daily living) and other aspects of quality of life is largely understudied in the MBC setting.

Studies included in our review that evaluated MBC patient preferences for avoiding treatment side-effects typically focused on avoiding short-term side-effects, such as nausea/vomiting, rather than long-term or severe and persistent side-effects that may have a greater impact on patient quality of life, including physical, mental, emotional, and social well-being. Preferences relating to quality of life versus length of life may also be influenced by other patient characteristics, such as older age. Older adults with cancer have been found to consistently prioritize quality of life over length of life [39,40,41]. Given that older adults may suffer from multiple comorbidities, they may not tolerate specific cancer treatments as well and have an increased risk of more severe treatment side effects and subsequent physical function declines [39]. Few older adults indicate that they are willing to trade certain aspects of quality of life, such as cognitive functioning and physical function, for survival [41,42]. Overall, there is a need to better understand the potential trade-offs that people with MBC are willing to make between quality of life and treatment efficacy and the reasons underpinning such decision-making. Future studies should seek to include quality of life as an outcome when assessing patient preferences, as well as among key subgroups, such as older versus younger adults. It is worthwhile to understand the relative importance of all dimensions of quality of life to better understand patients’ needs and priorities to tailor their cancer care, accordingly.

There were only four studies that explored patient preferences relating to communication and decision-making [31,32,34,35]. Yet, studies still focused on communication and decision-making preferences relating to cancer treatment and not multidimensional care. Two studies explored patient views and perspectives of communicating prognosis and reported that preferences relating to prognostic information communication and decision-making are nuanced. When patients are seeking prognostic information, it is rare they want simple or straightforward statistics on life expectancy [34]. In oncology, especially the advanced and metastatic cancer setting, HCPs are more than “a source of information” to patients [43]. Rather, patients may also look to their HCPs as sources of hope, guidance, understanding, and emotional support [43]. Findings from the current review suggest data on MBC patient communication, decision-making, and healthcare information preferences are limited. In the current review, we did not identify any quantitative research on communication and decision-making preferences. Additional quantitative data may provide important information from larger MBC patient samples and offer more generalizable evidence.

Observational cross-sectional studies in patients with mixed advanced cancer types have found patients often prefer receiving as much information as possible about their diagnosis, treatment, and prognosis [44] and prefer communication from HCPs that is realistic and individualized [45]. However, findings from mixed cancer studies may not directly extend to people with MBC due to differences in prognosis, treatment approaches, and location of metastasis between different tumour types. Furthermore, information on communication and decision-making preferences surrounding aspects of MBC care beyond cancer treatment is needed. No study included in our review specifically explored communication or decision-making preferences for other components of MBC multidimensional care, for example, palliative or supportive care. Consequently, more quantitative and qualitative evidence on communication and decision-making preferences for all components of MBC multidimensional care is needed.

We identified three studies focused on exploring preferences relating to supportive care among people with MBC. People with MBC can experience multiple quality of life concerns, such as cancer symptom burden, psychological distress, body image disturbances, disrupted daily activities, and social constraints [46]. Quality of life concerns can be addressed through supportive care services [47]. However, we noted that research on patient preferences for supportive care remains limited. One qualitative study summarized preferences relating to self-management practices [36]. Two studies explored patient preferences relating to physical activity programming to help inform the design of future interventions for patients [29,33]. The benefits of physical activity have been widely demonstrated among patients with early-stage breast cancer [48,49,50]. Data are more limited in the MBC setting; however, evidence suggests that physical activity may help to address a number of quality of life concerns in people with advanced cancer [51,52,53]. Notably, the burden of cancer symptoms may act as a barrier to physical activity participation for some people with MBC, whereas for others, there may be significant interest in physical activity programming despite marked symptom burden [29,33]. As a result, more information on patients’ preferences regarding physical activity is crucial to tailor physical activity recommendations and programs appropriately. We also did not identify any studies that explored patient preferences for interventions targeting psychological health, such as pharmaceutical treatments, psychotherapy, or supportive–expressive group therapy. People with MBC frequently experience cancer-related distress [7], which can be highest at the time of diagnosis [54]. Greater anxiety and depression may also be associated with higher physical symptoms scores [55,56]. Thus, there is a need to better understand patient preferences for psychological health interventions, including the type, timing, and relative importance to other supportive care services to improve the comprehensive care of people with MBC.

### 4.1. Study Limitations

The current scoping review includes broad inclusion criteria to comprehensively map out the scientific evidence on the topic of MBC patient preferences. We included studies reporting patient preferences regarding all aspects of care to scope the body of literature, identify key knowledge gaps, and elucidate important areas for future research. We also conducted a rigorous and transparent methodological process, including a comprehensive search strategy, and independent article screening and review. Because the current review is a scoping review and included heterogeneous studies, no analysis beyond a narrative description of the included studies was performed. Scoping reviews provide a broad overview of a topic, but do not provide an in-depth analysis on the summarized data. However, our findings may serve as a precursor for future systematic reviews on relevant topics relating to MBC patient preferences. Other limitations of scoping reviews include the lack of risk of bias assessment, which impacts our ability to determine study quality.

### 4.2. Future Directions

People with MBC may be presented with numerous complex care-related decisions over the course of their cancer treatment trajectory. Many decisions are sensitive to the preferences of patients, particularly if prognosis or treatment outcomes are uncertain and available treatments may compromise quality of life. HCPs are often unable to accurately predict patient preferences [12,57]. HCPs may also not be inclined to or possess the necessary clinical communication skills to educate patients on their care options and choices [58,59]. Some HCPs may unintentionally overestimate benefits, while minimizing harms of treatments [60], or steer patients in certain directions [61]. Further research on the care preferences of people with MBC is thus needed to inform HCPs on what is most valuable to patients and improve the delivery of MBC care.

To better understand the preferences of people with MBC and improve MBC care, our review has identified important topics for future research. Firstly, research on treatment preferences has been restricted to tumour-directed therapies. There remains a lack of research exploring patient preferences for MBC medical treatments for the management of symptoms arising from cancer and tumour-directed treatments. People with MBC often experience unique and more burdensome symptoms relative to patients with early-stage breast cancer, such as cancer-related distress and pain, shortness of breath or persistent coughing, and neurological symptoms due to metastases to the bones, lungs, and brain. Research on patient treatment preferences for the management of MBC symptoms would be worthwhile to help optimize patient care and quality of life. Next, there is a specific need to further investigate patient preferences for non-medical interventions as a part of multidimensional MBC care, including supportive care services across multiple disciplines (e.g., psychological care, spiritual and social supports, pain management, physical rehabilitation, communication preferences, patient advocacy, bereavement support and end of life planning) [62]. Supportive care can help patients and their families cope with a cancer diagnosis and adapt to cancer treatment regimens to maximize treatment efficacy and minimize its burdens [62]. Understanding patients’ decision-making process in terms of making trade-offs to engage with supportive care is an important next step. More research on the optimal timing for delivery of supportive care services for MBC is also needed [63]. Lastly, we found that the views of HCPs were typically only included in studies that focused on preferences for MBC tumour-directed treatment. Evaluating HCP preferences regarding other aspects of MBC multidimensional care may also help identify potential mismatches to patient preferences, gaps in clinical practice, and clinical decision-making patterns on discussions and referrals to all available MBC services and interventions.

## 5. Conclusions

The current scoping review summarizes the evidence on the preferences of people with MBC regarding all aspects of their cancer care. Overall, our findings indicate that most studies to-date have evaluated MBC patient preferences for tumour-directed cancer treatments. Information on patient preferences relating to other aspects of MBC care is more limited, including supportive care services that may be accessed at various stages of patients’ care trajectories. People with MBC are living longer with their cancer and frequently require multidisciplinary, multi-modal care to manage both physical and psychosocial health and well-being. Consequently, there is a need to understand MBC patients’ preferences for care beyond tumour-directed cancer treatments to provide a more complete picture of what is most important to patients throughout their entire cancer care experience.

## Figures and Tables

**Figure 1 cancers-15-04331-f001:**
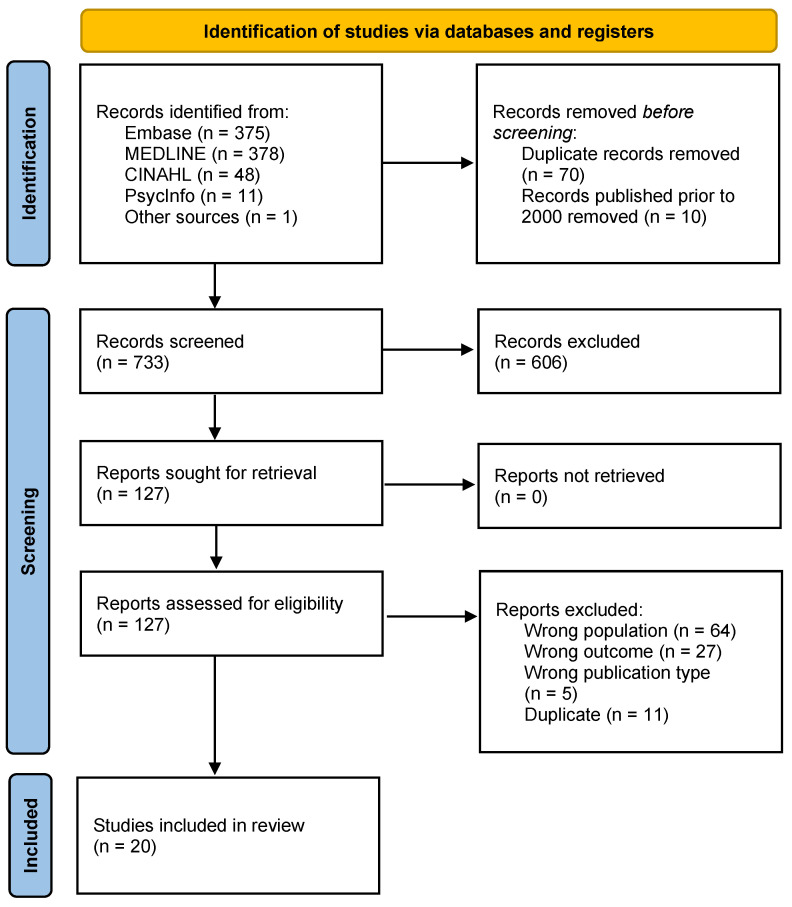
Flow diagram.

**Table 1 cancers-15-04331-t001:** Studies reporting on patient preferences for metastatic breast cancer care.

Study	Country	Study Design	Sample Size	Participants	Mean Age (±SD or Range) ^1^	De Novo Status	Preference Type	Outcome Measures *	Follow-Up	Results *
**Treatment Preferences**										
Specific Drug Preferences										
Decker et al., 2020 [17]	Germany	RCT	*n* = 192 (Patients) *n* = 13 (HCPs)	Post-menopausal women with HR+/HER2− locally recurrent and inoperable or metastatic breast cancer Physicians	Arm A: 64.4 (47–83.6) Arm B: 65.9 (49.8–86.0)	NR	Preferences for combined antihormonal therapy (everolimus + exemestane) or chemo- and anti-angiogenic therapy (capecitabine + bevacizumab)	PPQ	3 mo; 24 mo	Patients and healthcare providers tended to prefer capecitabine + bevacizumab.
Treatment Administration Preferences										
Ciruelos et al., 2020 [18]	Spain	RCT	*n* = 166 (Patients) *n* = 39 (HCPs)	Women with HER2+ metastatic breast cancer Oncology nurses, medical oncologists, general nurses, others	60 (35–93)	De novo: *n* = 61 (36.7%) Recurrence: *n* = 105 (63.3%)	Preferences for subcutaneous versus intravenous trastuzumab administration	Researcher-developed survey	After 2 cycles; after 4 cycles	Most patients and healthcare providers tended to prefer subcutaneous trastuzumab.
Fallowfield et al., 2011 [30]	UK	Mixed methods	*n* = 79	Women with breast cancer with bone metastases	Oral Group: 62.3 ± 11.9 IV Group: 62.6 ± 13.2	NR	Preferences for oral versus intravenous bisphosphonates administration	Semi-structured interviews	3 mo; 6 mo	Both oral and intravenous bisphosphonates had disadvantages but were acceptable to most patients.
Gornas and Szczylik 2010 [28]	Poland	Cross-sectional observational	*n* = 215	Women with metastatic breast cancer	52 (27–77)	NR	Preferences for oral versus intravenous capecitabine administration	Researcher-developed survey	None	Most patients tended to prefer oral chemotherapy due to increased convenience and the possibility of staying at home during treatment.
Pivot et al., 2017 [19]	France	RCT	*n* = 113	Patients with HER2+ metastatic breast cancer	59.4 (34.7–84.9)	De novo: *n* = 58 (51.3%) Recurrence: *n* = 55 (48.7%)	Preferences for subcutaneous versus intravenous trastuzumab administration	PPQ	After 3 cycles	Most patients tended to prefer subcutaneous trastuzumab.
Treatment Characteristic Preferences										
Amin et al., 2022 [20]	USA	DCE	*n* = 169 (Patients) *n* = 117 (HCPs)	Patients with locally advanced or metastatic triple-negative breast cancer or endocrine refractory HR+ breast cancer Oncologists	54.2 ± 9.2	NR	HER2- treatment preferences (OS, PFS, neuropathy, neutropenia, nausea, alopecia, immune-related AEs)	Researcher-developed surveys	None	Improving OS was most important to patients and HCPs, followed by improving nausea and neuropathy.
DaCosta DiBonaventura et al., 2014 [23]	USA	Conjoint Analysis	*n* = 181	Women with metastatic breast cancer	52.2 ± 9.1	NR	Treatment preferences (OS, quality of life, treatment side-effects, dosing regime) and trade-offs between treatment side effects and effectiveness/quality of life	Interviews to develop survey (*n* = 10) Researcher-developed survey	None	Improving OS was most important to patients, followed by improving alopecia, fatigue, neutropenia, neuropathy and nausea/vomiting.
Lalla et al., 2014 [24]	USA	Conjoint Analysis	*n* = 298	Patients with metastatic breast cancer	<30 to 71+	NR	Treatment preferences and willingness to pay to avoid treatment side effects	Researcher-developed survey	None	Patients were willing to pay the most to avoid severe diarrhea, followed by being hospitalized due to infection, severe nausea and tingling in hands and feet.
Maculaitis et al., 2020 [21]	USA	DCE	*n* = 513 (Patients) *n* = 209 (HCPs)	Postmenopausal women with HR+/HER2- metastatic breast cancer Medical oncologists	47.4 ± 9.9	NR	CDK4/6 inhibitor treatment preferences (dose reduction, treatment side effects, dose regimen, dose schedule) and trade-offs between treatment benefits and risks	Interviews to develop survey (patients, *n* = 10; oncologists *n* = 8) Researcher-developed survey	None	Avoiding diarrhoea and Grade 3–4 neutropenia were of most importance to patients and oncologists.
Mansfield et al., 2022 [22]	USA, UK Japan	DCE	*n* = 302	Patients with advanced or metastatic breast cancer	47.6 ± 11.5	NR	HER2- treatment preferences (PFS, treatment side-effects) and trade-offs between treatment benefits and risk	Researcher-developed survey	None	Improving PFS was most important to patients, followed by reducing the risk of heart failure.
Smith et al., 2014 [25]	USA	Conjoint Analysis	*n* = 641	Patients with metastatic breast cancer	40–80+	NR	Paclitaxel and capecitabine preferences (benefit, treatment side effects)	Researcher-developed survey	None	Treatment benefit was more important than treatment side effects to patients.
Spaich et al., 2018 [26]	Germany	Conjoint Analysis	*n* = 100	Patients with metastatic breast cancer	64.4 ± 10.6	NR	Taxane chemotherapy preferences (PFS, application time, cycle, premedication, treatment side effects)	Researcher-developed survey	None	Avoiding Grade 3–4 neutropenia was most important to patients, followed by alopecia, Grade 2–4 neuropathy and PFS.
Reinisch et al., 2021 [27]	Germany	Conjoint Analysis	*n* = 104	Postmenopausal women with HR+/HER2- locally advanced or metastatic breast cancer	50–70+	Recurrence: *n* = 72 (69%) De novo: *n* = 32 (31%)	Palliative treatment preferences and importance of OS/PFS relative to quality of life/treatment side-effects	Interviews to develop survey (*n* = 12) Researcher-developed survey	None	Improving quality of life (physical agility and mobility) was most important to patients, followed by OS and PFS.
**Communication and Decision-making Preferences**										
Butow et al., 2002 [34]	Australia	Qualitative	*n* = 17 (Patients) *n* = 13 (HCPs)	Women with metastatic breast cancer Oncologists, nurses, psychiatrist, psychologist, social worker, breast cancer advocate	50 (38–80)	NR	Views towards disclosing prognosis and the ideal manner in which to structure the discussion	Semi-structured interviews	None	Open and repeated negotiations for patient preferences for information. Patients tended to prefer aclear, straight-forward presentation of prognosis.
Ejem et al., 2018 [31]	USA	Mixed methods	*n* = 22	Women with metastatic breast cancer	62 (33–87)	NR	Treatment decision-making preferences (“shared” versus “independent” versus “delegated” decision making)	CPS Semi-structured interviews	3 mo	Patients selected a “shared” treatment decision-making style using the CPS. Interview descriptions reflected a passive process where patients followed oncologists’ treatment suggestions.
Niranjan et al., 2020 [35]	USA	Qualitative	*n* = 44 (Patients) *n* = 34 (HCPs)	Women with metastatic breast cancer Oncologists, nurses, lay investigators	50% of patients were 55 and over	NR	Communication preferences regarding prognosis, crisis support, treatment information, and timing of communication	Interviews (Patients) Focus Groups (HCPs)	None	Most patients expressed wanting prognostic information but varied in the timing of when they wanted the information.
Rocque et al., 2019 [32]	USA	Mixed methods	*n* = 20 (Patients) *n* = 11 (HCPs)	Women with metastatic breast cancer Community oncologists, academic oncologists	25–65+	NR	Factors influencing decision-making in treatment selection	CPS Interviews (Patients) Focus Group or Interviews (HCPs)	None	Patients and HCPs consider treatment characteristics when making decisions. Patients tend to have broader considerations than HCPs and incorporate more contextual factors.
**Supportive Care Preferences**										
Delrieu et al., 2020 [29]	France	Single-arm intervention trial	*n* = 49	Women with metastatic breast cancer	55 ± 10.4	De novo: *n* = 14 (28.6%), Recurrence: *n* = 35 (71.4%)	Physical activity preferences	Researcher-developed survey	6 mo	Physical activity preferences varied. Most patients tended to prefer receiving counselling from a physical therapist specialist, and preferred exercise during treatment, in the company of others and at home (baseline) or in a fitness centre (6 mo).
Schulman-Green et al., 2011 [36]	USA	Qualitative	*n* = 15	Women with metastatic breast cancer	52 (37–91)	NR	Self-management preferences, practices, and experiences	Semi-structured interviews	None	Self-managed preferences vary. HCPs should repeatedly explore patients’ self-management preferences and ability to self-manage.
ten Tusscher et al., 2019 [33]	The Netherlands	Mixed methods	*n* = 114	Patients with metastatic breast cancer	63.5 ± 10.2	NR	Exercise-based physical therapy program preferences	Researcher-developed survey Focus groups (*n* = 6)	None	Exercise-based physical therapy program preferences vary. Patients tend to prefer high-quality, physical therapist-guided, tailored exercise programs.

* Relating to patient preferences ^1^ Unless otherwise reported. AE: adverse events, CPS: Control Preference Scale DCE: Discrete Choice Experiment, HCPs: Healthcare Providers, OS: Overall survival, PASE: Physical Activity Scale for Elderly, PCS: Patient Specifics Complaints Instrument, PFS: Progression-free survival, PPQ: Patient Preference Questionnaire.

## Data Availability

The data can be shared upon request.

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
