# Peer review of "Patient Preferences in Metastatic Breast Cancer Care: A Scoping Review"

_cancers, 2023, doi:10.3390/cancers15174331_

Round 1

Reviewer 1 Report

Despite tremendous advances in breast cancer (BC) treatment, metastatic breast cancer (MBC) is usually the catastrophic turning point in a woman's life. Recognizing and understanding patient preferences is a critical step in promoting patient-centered care. So, I appreciate the work of Bland et al. submitted to “Cancers”. As the authors choose the format of “scoping review”, its primary purpose should be to focus on the selected research questions and provide evidence-based, comprehensive answers. Unfortunately, the work in its current form is rather a “soft” or “inferior” variant of a systematic review. What I suggest to improve is the art of presenting the results: The three paragraphs (3.1.-3.3.) are currently a simple juxtaposition of the identified studies. I think that each paragraph should represent the state of the art for the question being evaluated. Furthermore, the conclusions of this "scoping review" does not in fact provide any useful insight other than that the issue is important and that “recognizing and understanding patient preferences is a critical step in advancing patient-centered care”. Please summarize the conclusions in a systematic way, e.g. in the form of a table.

Author Response

Thank you for taking the time to review our manuscript and for providing this insightful feedback. The primary aim of our scoping review is to “map the available quantitative and qualitative evidence reporting on patient preferences in metastatic breast cancer (MBC) care and identify relevant knowledge gaps to inform directions for future research.” Unlike a systematic review, which is guided by a more specific research question and typically compares empirical evidence from often a smaller number of studies, a scoping review provides an overview of a larger and more diverse body of evidence. Our review captures quantitative and qualitative study designs on patient preferences for all aspects of MBC care. Thus, we feel a scoping review is most appropriate given the heterogeneity of studies. We do not feel our review is necessarily “softer” or “inferior” to a systematic review but rather, more closely aligns with our proposed research question.

Based on your feedback, we have gone through our results section to ensure that what we are reporting is in line with our primary aim. This includes revising sections 3.1-3.3 with statements that more clearly reflect what has been published to-date on MBC patient preferences (i.e., typically preferences related to treatment) versus what has understudied (i.e., other aspects of multidimensional MBC care). We have also reviewed the requirements of reporting results for a scoping review based on recommendations from the Joanna Briggs Institute. This includes providing a narrative summary that accompanies our tabulated results and describes how the results relate to our review’s aim.

In our “Future Directions” section of the discussion, we highlighted that patient preference research is needed because healthcare providers are often unable to accurately predict patient preferences. Consequently, further research on the preferences of people with MBC can help to inform healthcare providers on what is most valuable to patients and improve the delivery of MBC care. We have revised this section to also more systematically describe key conclusions and areas for future research. This includes more research on 1) patient preferences for MBC treatments beyond tumour-directed treatments, including the pharmaceutical management of cancer symptoms, 2) patient preferences for non-medical interventions, such as supportive care services and 3) healthcare provider preferences on aspects of MBC care outside of tumour-directed treatments to identify any misalignment with patient preferences and gaps in clinical practice.

Reviewer 2 Report

In my opinion, the analyzed topic is interesting enough to attract the readers’ attention.I think that the abstract of this article is very clear and well structured. In my opinion, the discussion could be studied in depth and extended. Maybe, it could be useful the evaluation of the newest treatments for complications as GMS in order to complete the discussion. In particular, I suggest these articles to get deeper in the topic: PMID: 36037664 and PMID: 37076125 . Because of these reasons, the article should be revised and completed. Considered all these points, I think it could be of interest for the readers and, in my opinion, it deserves the priority to be published after revisions.

minor editing is necessary

Author Response

Thank you for your review of our manuscript and for your recommendations for improvement. We have attempted to significantly revise our discussion to expand on key themes, while maintain focus on our research question. This includes expanding on gaps in patient treatment preferences research (i.e., focus on tumour-directed treatment preferences over preferences for other treatments) in paragraph 2 of the discussion. We now also highlight that studies on patient communication and decision-making preferences have again, typically only focused on preferences relating to receiving information about treatment (paragraph 4) and more information on communication preferences for other aspects of MBC care is needed. We then go on to elaborate on gaps in research on supportive care preferences among people with MBC (paragraph 5). In particular, we have noted there are no studies on patient preferences for psychological interventions to manage cancer-related distress among people with MBC. Lastly, we have significantly revised our “Future Directions” section of the discussion to more clearly articulate directions for future research. This includes more research on 1) patient preferences for MBC treatments beyond tumour-directed treatments, including the pharmaceutical management of cancer symptoms, 2) patient preferences for non-medical interventions, such as supportive care services and 3) healthcare provider preferences on aspects of MBC care outside of tumour-directed treatments to identify any mismisalignment with patient preferences and gaps in clinical practice.

Round 2

Reviewer 2 Report

The quality of the manuscript has improved thanks to the changes made. I think it could be of interest to the readers and, in my opinion, it deserves the priority to be published.